# Natamycin Has an Inhibitory Effect on *Neofusicoccum parvum*, the Pathogen of Chestnuts

**DOI:** 10.3390/molecules28093707

**Published:** 2023-04-25

**Authors:** Lin-Jing Gou, Tian-Tian Liu, Qi Zeng, Wan-Rong Dong, Lu Wang, Sha Long, Jiang-Tao Su, Yu-Xin Chen, Gao Zhou

**Affiliations:** 1Hubei Key Laboratory of Industrial Microbiology, Key Laboratory of Fermentation Engineering (Ministry of Education), Cooperative Innovation Center of Industrial Fermentation (Ministry of Education & Hubei Province), Hubei University of Technology, Wuhan 430068, China; 2National “111” Center for Cellular Regulation and Molecular Pharmaceutics, School of Biological Engineering and Food, Hubei University of Technology, Wuhan 430068, China; 3Post-Doctoral Research Center of Mayinglong Pharmaceutical Group Co., Ltd., Wuhan 430064, China

**Keywords:** natamycin, *Neofusicoccum parvum*, chestnut, antifungal mechanism

## Abstract

This research aimed to investigate natamycin’s antifungal effect and its mechanism against the chestnut pathogen *Neofusicoccum parvum*. Natamycin’s inhibitory effects on *N. parvum* were investigated using a drug-containing plate culture method and an in vivo assay in chestnuts and shell buckets. The antifungal mechanism of action of natamycin on *N. parvum* was investigated by conducting staining experiments of the fungal cell wall and cell membrane. Natamycin had a minimum inhibitory concentration (MIC) of 100 μg/mL and a minimum fungicidal concentration (MFC) of 200 μg/mL against *N. parvum*. At five times the MFC, natamycin had a strong antifungal effect on chestnuts in vivo, and it effectively reduced morbidity and extended the storage period. The cell membrane was the primary target of natamycin action against *N. parvum*. Natamycin inhibits ergosterol synthesis, disrupts cell membranes, and causes intracellular protein, nucleic acid, and other macromolecule leakages. Furthermore, natamycin can cause oxidative damage to the fungus, as evidenced by decreased superoxide dismutase and catalase enzyme activity. Natamycin exerts a strong antifungal effect on the pathogenic fungus *N. parvum* from chestnuts, mainly through the disruption of fungal cell membranes.

## 1. Introduction

Chestnut (*Castanea mollissima* Blume) belongs to the genus *Castanea* and is one of the most important cash crops in China. Chestnut represents both the chestnut tree and the chestnut fruit, and, in this study, chestnut refers to the chestnut fruit. Chestnut fruits have high edible value, and their abundant nutritional elements have long been established [1]. Chestnut starch has a high concentration of resistant starches, which increase satiety and energy maintenance by extending the time spent in the digestive tract [2]. Furthermore, chestnut fruits are rich in protein, vitamins, fiber, essential fatty acids, and minerals, as well as polyphenols with antioxidant activity [3]. On the other hand, chestnut fruits exhibit medicinal value because of their multiple biological activities. According to Rodrigues et al. [4], mice whose diets include chestnut fruits accumulate low belly fat and have low serum cholesterol levels. Chestnut starch has probiotic activity and can be used to improve human gastrointestinal health by synthesizing isomalto-oligosaccharides through various pathways, thereby promoting *Lactobacillus proliferation* [5]. Chestnuts have an inhibitory effect on *E. coli*, whether they are digested in vitro or not [6]. Furthermore, extracts of chestnut pellicle can reduce scab disease caused by *Streptomyces scabies* [7]. A water-soluble polysaccharide derived from chestnut fruit can also cause human cervical cancer cells to undergo apoptosis via the mitochondrial route [8]. The oxidative stress caused by H_2_O_2_ and dextran sulfate sodium salt on IPEC-J2 cells was greatly reduced, and the cellular activity was increased via pretreatment with the chestnut–Quebracho mixture for 3 h [9].

However, chestnut fruits are prone to rot and deterioration during the harvesting and storage periods because of their high water content, thus seriously affecting their quality and production value [10]. Fungal infestation is a major cause of chestnut fruit decay during storage, and Waqas et al. [11] first reported *Neofusicoccum parvum* as the pathogenic fungus that causes nut rot in Italian hazelnuts. Italian scholars Seddaiu et al. [12] have isolated and identified *N. parvum* from rotting chestnut fruits.

*N. parvum* is a plant pathogenic fungus belonging to the Botryosphaeriaceae [13,14]. *N. parvum* is distributed worldwide and is a major source of infestation in various plants. *N. parvum* is the primary pathogen of Chinese gallnut brown spot; sequoia canker and dieback; Scaevola taccada leaf spot; and avocado branch cankers [15,16,17,18]. *N. parvum* acts as an endophytic fungus that can invade plant tissues and cells through wounds caused by cutting branches, thus weakening the tree and, in severe cases, leading to plant death [19,20]. However, the mechanism of infection of plant hosts by *N. parvum* infestation is not clear, and some studies suggest that the pathogenicity of *N. parvum* may be related to its ability to colonize plant tissues, the production of toxins, and the production of extracellular proteins that have toxic effects on plants [19]. Accordingly, pathogenic infestation needs to be suppressed to reduce plant diseases and increase agricultural product yield.

Natamycin (NM) is a natural antifungal polyolefin macrolide that is internationally licensed as a microbial-derived antiseptic [21]. This type of preservative has the advantages of being environmentally friendly, safe, and effective, and it is widely used in food preservation [22]. The national standard of the People’s Republic of China (GB 2760-2014) clearly stipulates that the maximum use level of natamycin in food is 0.3 g/kg and the maximum surface residue does not exceed 10 mg/kg [23].Galotyri cheese may be effectively protected from fungal growth while being stored using natamycin, thus extending the cheese’s shelf life [24]. Guo et al. [25] discovered that, by regulating the antioxidant enzyme activity of green-skinned walnuts, natamycin could effectively inhibit the increase in the mold rot rate. Zhou et al. [26] applied a film treatment to red globe grapes and discovered that natamycin could effectively reduce the fruit’s respiratory intensity and decay rate while significantly extending the fruit’s storage period. Natamycin’s antifungal mechanism is related to its molecular structure, in which the hydrophobic macrolide double bond can bind to sterol molecules in the fungal cytoplasmic membrane, thus increasing the cytoplasmic membrane permeability. The hydrophilic macrocyclic lactone polyol fraction, on the other hand, enhances membrane permeability by creating water pores in the membrane, allowing macromolecules such as proteins and nucleic acids to flow out of fungal cells [27].

Accordingly, this research investigated the antifungal activity of natamycin against *N. parvum*, by conducting in vivo and in vitro experiments on chestnut fruits, and further explored its antifungal mechanism of action to provide the scientific basis for the application of natamycin in chestnut preservation.

## 2. Results and Analysis

### 2.1. Growth Curve of N. parvum

*N. parvum* mycelium grows very quickly, and it took only 4 days to occupy a 9 cm-diameter plate of PDA medium at a constant temperature of 28 °C. At the beginning of growth, the mycelium was white. As the incubation time increased, the mycelium gradually changed to gray-green and finally to gray-black, from inside to outside. Natamycin treatment remarkably inhibited mycelial growth, as shown in Figure 1, and the inhibitory effect increased with increasing concentration. At the concentration of 200 μg/mL, the inhibition rate reached 100% each day, indicating that *N. parvum* growth was completely inhibited at this concentration. Therefore, 200 μg/mL natamycin can be considered as the lowest concentration (MFC) that can kill *N. parvum*. *N. parvum* mycelial growth was also inhibited by 10 and 50 μg/mL natamycin, with inhibition rates ranging from 47.66% ± 0.04% to 86.34% ± 0.01%. At the concentration of 100 μg/mL, the inhibition rate reached 100%, but the inhibition rate decreased with the extension of the observation time, indicating that this concentration was the MIC.

### 2.2. Inhibition Mechanism of Natamycin against N. parvum

#### 2.2.1. Effect of Natamycin on the Cell Wall of *N. parvum*

CFW can specifically bind chitin in the fungal cell wall and produce bright blue fluorescence under excitation light; accordingly, the damage of the fungal cell wall was determined based on the intensity of fluorescence [28]. AKP was present between the fungal cell membrane and the cell wall and can leak outside the cell only when the cell wall is disrupted. However, as shown in Figure 2, our results did not detect the effect of natamycin on the cell wall of *N. parvum*. After CFW staining, the control group mycelium showed bright blue fluorescence, while the 6.25, 12.5, and 25 μg/mL natamycin-treated groups showed the same bright fluorescence. In comparison with the AKP activity of the control group, no significant change was observed in the natamycin group at 1/2MIC and subinhibitory concentrations, indicating that natamycin did not cause AKP leakage from *N. parvum*.

#### 2.2.2. Effect of Natamycin on the Cell Membrane of *N. parvum*

Evans blue is a non-membrane-permeable dye that can be used to detect cell viability when the plasma membrane is damaged, and the dye can enter the cytoplasm and nucleus, thus staining them blue. As shown in Figure 3A, the mycelium of the natamycin-treated group was stained with Evans blue and visible under the microscope in a distinct blue color, while the mycelium of the control group showed no change. The staining deepened with the increase in natamycin action concentration. Therefore, natamycin caused damage to the cell membrane of *N. parvum*, and the degree of damage is positively correlated with the drug concentration.

The synthesis of ergosterol in the cell membrane of *N. parvum* was inhibited after treatment with natamycin, and the resulting statistics are shown in Figure 3B. The results show that 50 μg/mL natamycin inhibited ergosterol synthesis by 21.42% ± 0.02%, and, when the concentration was increased to 80 μg/mL, the inhibition rate reached 59.57% ± 0.06%. Therefore, different concentrations of natamycin can substantially inhibit ergosterol synthesis when compared with the control group, and when the inhibition rate is dose-dependent.

#### 2.2.3. Effect of Natamycin on the Leakage of the Cellular Contents of *N. parvum*

As shown in Figure 4, natamycin can cause intracellular nucleic acid and protein leakage in *N. parvum*, and the ratio of leakage increases with the administrated concentration.

#### 2.2.4. Effects of Natamycin on the Oxidative Stress Response of *N. parvum*

The changes in the SOD and CAT activities of *N. parvum*, after different concentrations of natamycin treatment, are shown in Figure 5. Natamycin has a strong inhibitory effect on SOD and CAT activity, and both parameters showed a significant decreasing trend as the natamycin concentration increased. The results show that natamycin could exert antifungal effects through oxidative damage.

### 2.3. In Vivo Experiments

#### 2.3.1. In Vivo Antifungal Efficacy of Natamycin

The experiment was carried out with the cutoff point of all chestnut kernels in the model group being infested with *N. parvum*, and Figure 6A depicts the degree of infestation in each group on the last day of the experiment. The control group remained consistent with the initial state, the degree of infestation by *N. parvum* in the model group of chestnut kernels was 100%, and the degree of the chestnut kernels’ decay varied at different doses of natamycin treatment groups. The incident rates of chestnut kernels in each group were counted, and the results are shown in Figure 6B. All of the chestnut kernels in the model group rotted, which was highly significant when compared with the control group. Among them, the incident rates were 40% and 27% in the low- and medium- dose groups, respectively, while natamycin in the high-dose group could completely inhibit the infestation of chestnut kernels by *N. parvum*. In contrast to the model group, natamycin emulsion at low, medium, and high doses showed highly significant inhibition of *N. parvum* infestation, and the inhibition rate increased with the increase in concentration.

#### 2.3.2. Effect of Natamycin Emulsion on Postharvest Ripening Infection of the Chestnut Shell Bucket

The above experiments have confirmed the effective antifungal effect of natamycin on chestnut kernels and further expanded the experiments to study the antifungal effect of natamycin on chestnut shell buckets after harvesting. As shown in Figure 7, the decay rates of chestnuts were counted based on the number and weight of decayed fruits. Both doses of natamycin emulsion inhibited the decay of post-ripe chestnuts during storage, while the decay rate in the five times MFC group was lower than that in the MFC group, indicating that the preservative effect of the natamycin emulsion was positively correlated with its dose. The natamycin emulsion could effectively reduce the decay rate of chestnuts during storage, improve the quality of chestnuts, and prolong the storage period.

## 3. Discussion

Natamycin is the only antifungal additive approved by the Food and Drug Administration in the United States [29] and is widely used in the food preservation industry, such as the preservation of dairy products, meat and meat products, fruits and vegetables, and bakery products [30]. This paper involves the determination of natamycin’s antifungal activity against *N. parvum*, the causative agent of chestnut fruit rot, and its antifungal mechanism. Based on the plot of the growth curve of *N. parvum*, the MIC and MFC of natamycin were 100 and 200 μg/mL, respectively. On this basis, the infestation experiments were carried out by reincorporating *N. parvum* into chestnuts, and the results show that natamycin could completely inhibit the growth of *N. parvum* at five times the concentration of MFC. Next, the scope of the experiment was expanded to outside the laboratory, and the natamycin emulsion was sprayed into the postharvest mature chestnut shell bucket. The results show that five times MFC natamycin could prevent chestnuts from rotting. However, experimental studies on the preservation of post-ripe chestnuts have some limitations. Considering that chestnuts carry a large number of microorganisms by themselves, and given the inability to guarantee the sterility of the experimental process, the chestnuts in the control group also showed a high rate of decay. Moreover, in addition to fungal infestation, the ripening stage of chestnut fruit is susceptible to many factors such as temperature, humidity, and shell bucket maturity, resulting in many uncertainties. According to the aforementioned findings, natamycin has a potent inhibitory effect on *N. parvum*. However, compared with in vitro application, the effective concentration applied to chestnuts in vivo was much higher. This finding was obtained possibly because the antifungal effect of natamycin in chestnuts is influenced by many factors, and the nutritional conditions of chestnuts themselves are more suitable for fungal growth compared with a PDA medium, causing some interference for natamycin. Panagou et al. [31] found that 50 mg/L of natamycin completely inhibited the growth of surface fungi on chestnut fruits. A comparison of experimental methods revealed that Panagou et al. used immersion administration, while we sprayed natamycin on the surface of the chestnut shell bucket. The difference in the natamycin application methods most likely led to the difference in results. In the in vivo study on chestnut kernels, the highest concentration of natamycin was 1000 μg/mL, the mass of chestnut kernel pieces was 1 g, the volume of application was 5%, and the final calculated dose of chestnut kernel pieces was 50 mg/kg. In the in vivo study on chestnut shell buckets, the highest concentration of natamycin was 1000 μg/mL, the volume of application was 2% per kg of chestnut shell buckets, and the final calculated dose of chestnut shell buckets was 20 mg/kg. It can be seen that the amount of natamycin applied to either the chestnut kernel or the chestnut shell bucket does not exceed the maximum use level specified in the national standard. Moreover, natamycin is only sprayed on the surface of the chestnut shell bucket, which is removed before eating and will not affect the inner chestnut fruit. Washing before eating and consumption during storage will also further reduce natamycin residues on the surface of the chestnuts. We will specifically design experimental protocols in subsequent experiments to determine the actual residues of natamycin on the surface of chestnuts.

On this basis, this experiment provides a preliminary exploration of the antifungal mechanisms of natamycin. Most of the current studies on antifungal mechanisms of action have focused on the fungal cell membrane, cell wall, and energy metabolism. Liu et al. [32] showed that dill seed essential oil (DSEO) could disrupt the cell wall of *N. parvum* and emit a weak blue fluorescence under CFW staining. However, in the present study, mycelium could still emit a bright blue fluorescence by CFW staining after natamycin treatment, which does not differ from the blank mycelium without natamycin action. Similarly, natamycin did not change the AKP activity of *N. parvum*. Therefore, natamycin has no destructive effect on the cell wall of *N. parvum*, and the cell wall is not its target of action. The different effects on the fungal cell wall may be related to the different types of antifungal drugs. Ergosterol is an important component of the eukaryotic cell membrane. Natamycin has a strong affinity for ergosterol on fungal cell membranes and can irreversibly bind to it to form polyene–sterol complexes [30]. Our experimental results showed that natamycin treatment remarkably decreased the ergosterol content but substantially increased the leakage of cell contents. Moreover, the mycelium of the natamycin group could be stained blue by Evans blue, and the higher the concentration, the darker the blue color, which formed a significant difference from the control group. Therefore, one of the targets of natamycin’s action on *N. parvum* is the cell membrane, which disrupts the fungal cell membrane by binding specifically to ergosterol, further causing the vital fungal substances to leak out and die. This result is consistent with the findings of Aparicio et al. [33].

The effect of natamycin on the oxidation reaction of *N. parvum* was also investigated. Various biological processes in organisms result in reactive oxygen species (ROS), which cause oxidative stress. In response to such oxidative stress, organisms can deploy SOD and CAT to scavenge ROS, in order to protect cellular homeostasis [34]. In the present study, natamycin treatment remarkably reduced the SOD and CAT enzyme activities in *N. parvum*, indicating that natamycin can cause oxidative damage in *N. parvum*.

The antifungal targets of natamycin on *N. parvum* mainly include the cell membrane and the antioxidant enzymes SOD and CAT, while the cell wall is not its target. However, the energy metabolism of the fungus has not been determined, which can be the next research direction.

## 4. Materials and Methods

### 4.1. Experimental Materials

After surface disinfection of the rotten chestnut samples with alcohol, the samples were cut with a sterile blade and a portion of the rotten pulp was transferred to a blank PDA plate and incubated at 28 °C. The pathogenic chestnuts were isolated and the mycelium at the edge of the colony was picked and transferred to a new PDA plate three times to obtain a single strain and recorded with a number, which was stored on slant medium at 4 °C. The strain was further identified by Zhou [32] as *N. parvum*, No. 20221022-6-1, based on ITS sequencing, combined with mycelial morphology and other characteristics. *N. parvum* was activated using a PDA medium, incubated at 28 °C, and stored at 4 °C. Chestnut shell buckets were harvested from Yutoushan Village, Yantianhe Town, Macheng City, Hubei Province. Nearly ripe chestnut shell buckets were harvested from the orchard, selected, and then stacked at room temperature for reservation, and the chestnut fruits were separated and counted after 10 days. The collected chestnut fruits were stored at room temperature and the process of rot development during storage was counted. Chestnut kernels were purchased commodities and originated from Luotian County, Hubei Province. Chestnut kernels that were intact and free from infestation damage were selected for the experiment, and the unused kernels were stored at 4 °C and used within their shelf life.

Natamycin (food-grade) was obtained from Yino Biotechnology Co., Ltd., Ningbo, Zhejiang, China. Natamycin solution was prepared by dissolving the requisite amount in Tween-20 (0.1%, *v*/*v*). Evans blue (CAS:314-13-6) was obtained from BioFroxx, Calcofluor White (CAS:4193-55-9) was obtained from Sigma, and all the other chemicals were analytic-grade. A PDA medium was prepared by boiling 200 g potato, 18 g agar, and 20 g glucose with 1000 mL of distilled water. Unlike the PDA medium, the PDB medium does not require the addition of agar.

### 4.2. Effect of Natamycin on the Growth Diameter of N. parvum

The effect of natamycin on the mycelial growth of *N. parvum* was tested in vitro by using the agar dilution method [35]. Briefly, the natamycin stock solution was added to sterilized PDA medium to make final concentrations of 10, 50, 100, and 200 μg/mL. The medium containing different concentrations of natamycin (30 mL) was equally distributed to three plates and allowed to solidify. PDA medium without natamycin addition was used as a control group.

A 5 mm sterile punch was used to punch holes at the edges of actively growing colonies, and the mycelium was placed at the center of PDA plates containing different concentrations of natamycin, with the mycelium facing downward. Each treatment was performed in triplicates and the plates were sealed with sealant and incubated for 3 days upside down in a biochemical incubator set at a constant temperature of 28 °C ± 2 °C. Every 24 h, the growth of *N. parvum* was monitored. The crossover method was used to calculate the mycelial growth diameter of each group of plates and plot the *N. parvum* growth curve.

### 4.3. Inhibition Mechanism of Natamycin against N. parvum

#### 4.3.1. Effect of Natamycin on the Cell Wall of *N. parvum*

Calcofluor white (CFW) staining of *N. parvum* was carried out as described by Ouyang et al. [28]. An amount of 100 μL of the fungal suspension, at a concentration of 1 mg/mL, was added to a shake flask containing 25 mL of PDB medium and incubated in a shaker at 28 °C ± 2 °C and 200 rpm for 12 h. Various volumes of the stock solution of natamycin emulsion were pipetted at a concentration of 25 mg/mL into the conical flask to achieve final concentrations of 6.25, 12.5, and 25 μg/mL, which correspond to 1/16, 1/8, and ¼ minimum inhibitory concentration (MIC), respectively. The control group was not added with natamycin emulsion. After administration, the incubation was continued for 3 h, the mycelium was collected, and 10 μL of CFW and 10 μL of KOH (10%) were added dropwise and stained under dark conditions for 5 min. Excess dye was removed and observed via confocal laser scanning microscopy (Leica TCS SP8 CARS).

Fungal blocks were punched from PDA plates, placed in PDB medium containing natamycin emulsion at concentrations of 0, 50, and 80 μg/mL, and incubated in a shaker at 28 °C ± 2 °C and 200 rpm for 48 h. The supernatant was collected after centrifugation at 10,000 rpm for 10 min, the alkaline phosphatase (AKP) activity was measured strictly according to the AKP kit (Nanjing Jiancheng Bioengineering Institute, Nanjing, China) instructions, and the experiment was repeated thrice for each treatment.

#### 4.3.2. Effect of Natamycin on the Cell Membrane of *N. parvum*

Fungal cell membrane damage was observed by Evans blue staining according to the method of Lucía S. Di Ciaccio et al. [36]. An amount of 100 μL of mycelial suspension at 1 mg/mL concentration was added to a conical flask containing 25 mL of PDB and incubated in a shaker at 28 °C ± 2 °C and 200 rpm for 12 h. Various volumes of the stock solution of 25 mg/mL natamycin emulsion were pipetted into the conical flask to achieve final concentrations of 6.25, 12.5, and 25 μg/mL, corresponding to 1/16, 1/8, and ¼ MIC, respectively. The control group was not added with natamycin emulsion. Incubation was continued for 3 h with shaking under the same conditions as before. The mycelia were placed on glass slides and stained with 0.5% Evans blue dye solution dropwise for 5 min. The excess dye solution was washed off with PBS, and the staining situation was observed under an optical microscope (Olympus CX23, Beijing, China).

#### 4.3.3. Effect of Natamycin on Ergosterol Synthesis in *N. parvum*

The content of ergosterol was determined by UV spectrophotometry, according to the method of Abhishek et al. [37]. Mycelial blocks were punched at the edge of the colony with a puncher, placed in a conical flask containing 25 mL of PDB, and incubated in a shaker at 28 °C ± 2 °C and 200 rpm for 48 h. The mycelium was homogenized with a homogenizer and then expanded to a sufficient amount. The cultured mycelial suspension was equally divided into 50 mL centrifuge tubes with 25 mL of sample per tube. Natamycin emulsion stock solution (25 μg/mL) was pipetted into a centrifuge tube to prepare final natamycin concentrations of 50 and 80 μg/mL (i.e., 1/2 MIC and subinhibitory concentration), respectively, and the samples were incubated with a shaker for 3 h.

Then, the samples were centrifuged at 4000 rpm for 10 min. Approximately 1 g of mycelium was weighed, added with 5 mL of 25% potassium hydroxide ethanol solution, shaken vigorously for 10 min, and placed in a water bath at 85 °C for 4 h. A mixture of 4 mL of water and n-heptane was added at a ratio of 1:3, shaken for 10 min, and allowed to stand at room temperature for stratification. The n-heptane layer was transferred to a 5 mL EP tube and stored at −20 °C for 24 h. A UV spectrophotometer was used for scanning at the full wavelength of 230−300 nm. Then, the ergocalciferol content was calculated according to the following equation:Dehydroergosterol % = (A230/518)/*w* × 100%(1)
Ergosterol % = (A282/290)/*w* × 100% − dehydroergosterol %(2)

In these equations, 518 and 290 are constants, and *w* is the mycelium wet weight.

#### 4.3.4. Effect of Natamycin on the Leakage of the Cellular Contents of *N. parvum*

The release of cell constituents into the supernatants was measured using a previously described method [38] with minor modifications. Mycelial blocks were punched at the edge of the colony with a puncher, placed in a conical flask containing 25 mL of PDB, and incubated in a shaker at 28 °C ± 2 °C and 200 rpm for 48 h. Then, various amounts of natamycin emulsion were added to achieve final concentrations of 50 and 80 μg/mL (i.e., 1/2 MIC and subinhibitory concentration). Then, incubation was continued for 3 h. The leakage of macromolecular compounds from *N. parvum* was ascertained by collecting the supernatant after centrifugation at 10,000 rpm for 10 min. The absorbance values were determined at 260 and 280 nm. Each parameter was tested in triplicate.

#### 4.3.5. Effect of Natamycin on Oxidative Stress in *N. parvum*

The fungus was cultured using the method under Section 4.3.4. Superoxide dismutase (SOD) activity and catalase (CAT) activity were measured using specific kits (Beyotime Biotechnology, Haimen, China).

### 4.4. In Vivo Assays

#### 4.4.1. In Vivo Antifungal Efficacy of Natamycin

The chestnut kernels were washed in sterile water and cut into small pieces of 1 g mass using a safety razor blade. Each chestnut kernel piece in the test group was inoculated with 50 µL of mycelial suspension at a concentration of 1 mg/mL, while the control group received no treatment. Each chestnut kernel piece in the low-, medium- and, high- dose groups was given 50 µL of natamycin at concentrations of 100, 200 and 1000 µg/mL, respectively, corresponding to MIC, MFC, and 5 times MFC of natamycin emulsion against *N. parvum*. Each group was given three plates with five chestnut kernel pieces on each plate. The samples were placed at room temperature, after being sealed with sealant, to observe the infection of the chestnut kernel pieces. The rate of chestnut kernel pieces rotting after *N. parvum* infestation was expressed as the incidence rate.

#### 4.4.2. Effect of Natamycin Emulsion on the Postharvest Ripening Infection of the Chestnut Shell Bucket

The natamycin emulsion consisted of 2% natamycin, 1% Tween 80, and 97% distilled water. Natamycin and Tween 80 were dissolved in distilled water and, after complete dissolution, they were thoroughly mixed via ultrasonication. In addition, untreated chestnut shell buckets were used as the control group. The prepared natamycin emulsion was stored at room temperature for 30 days without any delamination, indicating its good stability.

An amount of 5 mg of *N. parvum* mycelium was picked off from PDA medium with a sterile inoculating needle on an ultraclean bench, and added to 5 mL of PDB medium and homogenized to a 1 mg/mL suspension of mycelium. Initially, a 1 mg/mL suspension of *N. parvum* was sprayed on the chestnut shell bucket to cause chestnut pathogenesis, and this setup was considered as the model group. The model group of chestnut shell buckets was then sprayed with different concentrations of natamycin emulsion, and the decay was observed. Natamycin emulsions, diluted to MFC and MFC 5 times, were administered separately. The volume of mycelial suspension and natamycin emulsion sprayed was 2% of the weight of the chestnut shell bucket. The control group received no treatment. Each treatment group was assigned 30 chestnut shell buckets, and 3 replicates were set up for each treatment group. When the green color of the chestnut shell bucket completely disappeared, the bucket naturally cracked, and the shell of the chestnut fruit completely hardened and changed color, it was considered that the chestnut shell bucket had reached the ripe state. When the chestnuts ripened, the shell buckets were removed, and the chestnuts were observed every 4 days until the decay stabilized. The criteria for chestnut fruit decay are color change, loss of hardness, visible mycelial growth on the surface of some chestnut fruits, and a distinct smell of decay.

### 4.5. Statistical Analysis

Each group of experiments was repeated thrice, and the results were reported as the means ± SD. Analysis of differences between groups was evaluated via one-way analysis of variance based on Duncan’s post hoc test through the use of SPSS25.0 (SPSS, Chicago, IL, USA). Differences with *p* values less than 0.05 were considered significant.

## 5. Conclusions

In this paper, the antifungal effect of natamycin on *N. parvum* was investigated, and its mechanism of action was initially explored. The results showed that natamycin could disrupt the fungal cell membrane and make its contents leak. The action of natamycin can also cause oxidative damage to the fungus, resulting in impaired cell function. In vivo experiments showed that natamycin emulsion could effectively prolong the storage period of chestnuts and could be used as a new environmentally friendly preservative to prevent postharvest chestnut fruit rot.

## Figures and Tables

**Figure 1 molecules-28-03707-f001:**
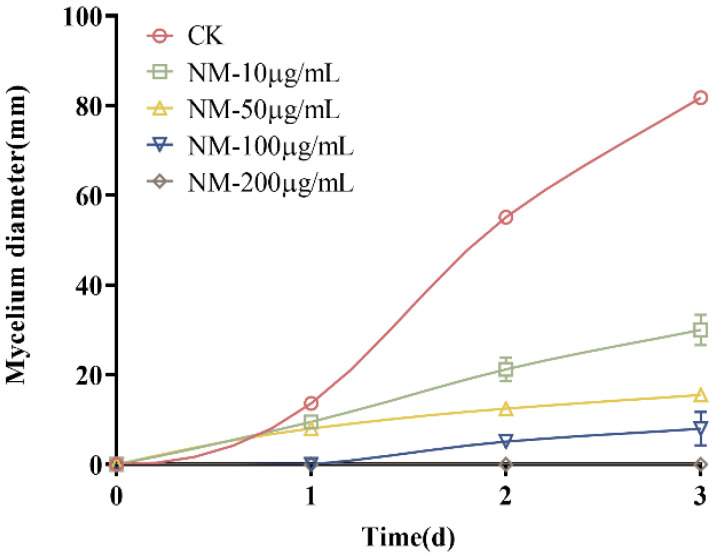
Inhibitory effects of natamycin on the growth of *N. parvum*.

**Figure 2 molecules-28-03707-f002:**
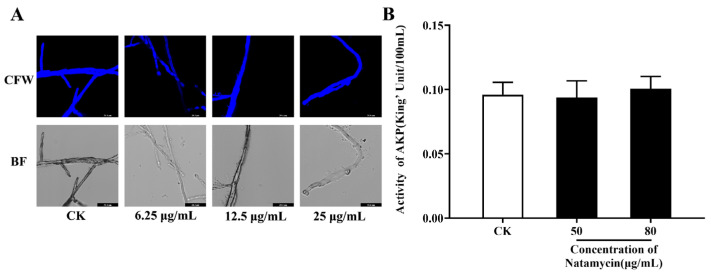
Effect of natamycin on the cell wall of *N. parvum*. Observation of mycelial changes after CFW staining under CLSM (**A**), and the effects of natamycin on the AKP activity of *N. parvum* (**B**), with 800× magnifications. Bar = 36.4 μm. “CK” stands for the control group.

**Figure 3 molecules-28-03707-f003:**
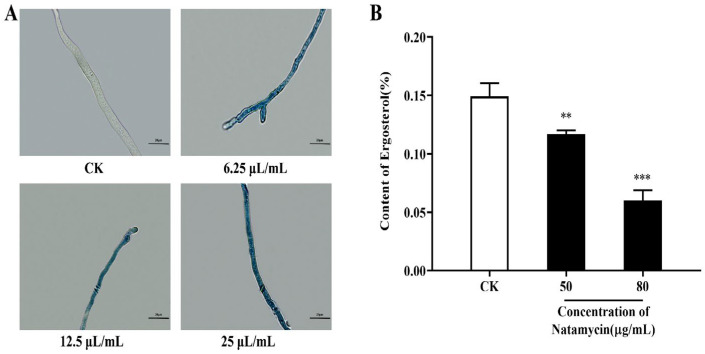
Effect of natamycin on the cell membrane of *N. parvum*. Observation of mycelial changes after Evans blue staining under the optical microscope (**A**), and the effects of natamycin on the ergosterol content of *N. parvum* (**B**), with 400× magnifications. Bar = 20 μm. “CK” stands for the control group; ** *p* < 0.01, *** *p* < 0.001 compared with the CK.

**Figure 4 molecules-28-03707-f004:**
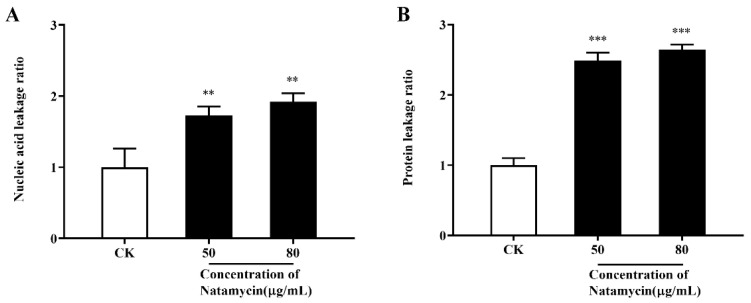
Effect of natamycin on leakage of cellular contents of *N. parvum*. Detection of nucleic acid leakage at 260 nm (**A**) and protein leakage at 280 nm (**B**). “CK” stands for the control group; ** *p* < 0.01, *** *p* < 0.001 compared with the CK.

**Figure 5 molecules-28-03707-f005:**
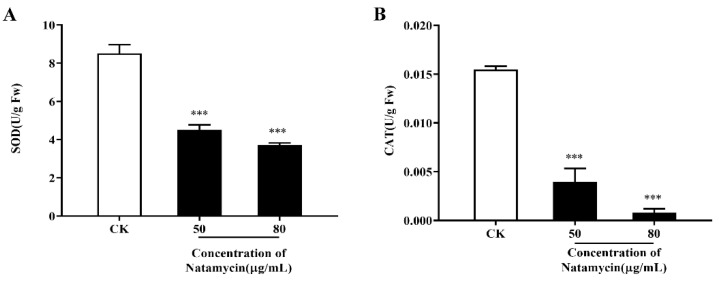
Effect of natamycin on oxidative-stress-related enzymes in *N. parvum*. The activity of SOD (**A**) and CAT (**B**). “CK” stands for the control group; *** *p* < 0.001 compared with the CK.

**Figure 6 molecules-28-03707-f006:**
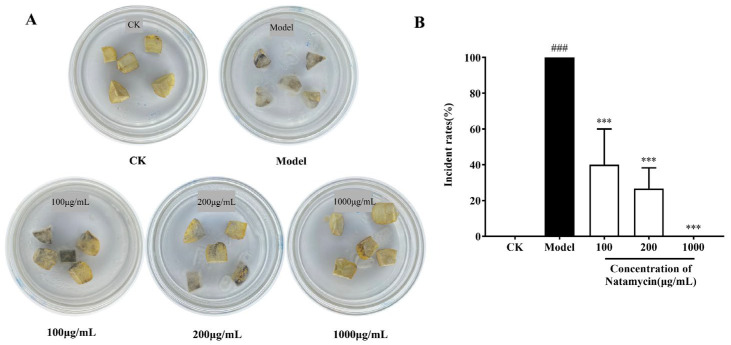
Effect of natamycin on the infestation of chestnut kernels by *N. parvum*. Representative photograph of natamycin inhibiting the growth of *N. parvum* in chestnut kernels (**A**), and the effects of natamycin on the incidence of chestnut kernels (**B**). All chestnut kernel pieces were incubated for 4 days. “CK” stands for the control group; “Model” represents the group of chestnut kernel pieces infected by *N. parvum*. “Incident rates” represent the rotting rates of *N. parvum*-infested chestnut kernel pieces in each treatment group. ^###^
*p* < 0.001 compared with the CK, *** *p* < 0.001 compared with the model group.

**Figure 7 molecules-28-03707-f007:**
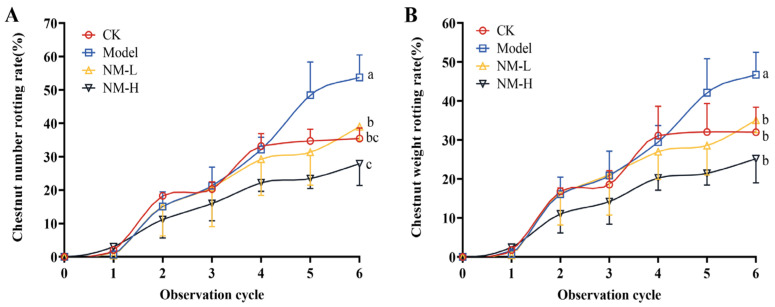
Effect of natamycin emulsion on postharvest ripening infection of chestnut shell bucket. Chestnut number rotting rate (**A**) and chestnut weight rotting rate (**B**). “CK”, control group; “NM-L”, 200 μg/mL natamycin emulsion; “NM-H”, 1000 μg/mL natamycin emulsion. Different letters (a, b, and c) indicate significant differences (*p* < 0.05).

## Data Availability

Data is unavailable due to privacy and ethical restrictions.

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
