# Peer review of "Natamycin Has an Inhibitory Effect on Neofusicoccum parvum, the Pathogen of Chestnuts"

_molecules, 2023, doi:10.3390/molecules28093707_

Round 1
Reviewer 1 Report
The manuscript describes an interesting experiment. Such treatments for postharvest storage of fruits or nuts is needed. Introduction, Materials and methods, and Results are very well and clearly written. However, there are some minor problems, mainly in the Discussion of the results.
Major comments:
In the Discussion, the results should be discussed with published papers describing the use of natamycin on similar products, like nuts, fruits or vegetables. This one describes natamycin on chestnuts https://doi.org/10.1080/14620316.2005.11511892, or different fruits https://doi.org/10.1016/j.postharvbio.2019.02.009.
Was some ANOVA post-hoc test used? If yes, provide details in 2.5.
Minor comments
L52 – full name of N. parvum should be given, as it is the first appearance in the main text.
L55 – the text „which was isolated from chestnut,“ could be deleted, it is already stated above
L93 – provide manufacturers of Evans blue and Calcofluor white
L346 - -give the full name of DSEO
Reviewer 2 Report
The subject of this research, improving treatments of chestnut kernels for commercial sale, is worthwhile. However, the authors do not describe their work in sufficient detail in places. Also they do not include an assessment of whether the levels of antifungal additive (natamycin) that provide some extension to good quality kernels are within levels permitted internationally in food. Their data also shows that there is some decay of the fruit despite their treatments, and this also needs to be discussed if the aim is to develop a treatment for commercial use.
There are some minor changes needed to the English, but the main problem with the manuscript is in the experimental design, results and discussion. To focus on the most significant:
Line 56, 57 The statement 'which are new species of asexual fungi' needs some clarification. Are there one, or several species?
Lines 55 - 63 Description of the fungus should include that it has been considered an endophyte, with implications of that for disease surveillance (See e.g. jof-08-00971-v2.pdf).
Lines 65 - 78 It would be useful to state the level of natamycin that is permitted in food (microgram per gram or another unit), since this is very relevant to this study.
Line 88 - 90 How were the chestnuts and chestnut kernels stored before experiments, and for how long?
Section 2.4.1 starts by mentioning chestnut kernels, but then only mentions chestnuts. Were the experiments performed on chestnuts, or chestnut kernels? Also, the description of the experimental protocol is not clear. Was each chestnut piece infected with 0.05 g of the fungal mycelium? At line 183 does it mean 5 whole chestnuts, or 5 x 1 g pieces of chestnut were on each plate? Please make this section clearer.
In Section 2.4.2 , Line 193 How was the dry mycelium prepared? Please refer to relevant earlier section or describe here. At line 200, how was ripening of the chestnuts identified? Also, state the number of chestnut bucket shells in each experimental group, and the number of replicates of groups. How was decay assessed?
Lines 212 - 213 Did the colour of the mycelial hyphae change, or was the color change due to formation of spores? Was there any difference in color change between the control and natamycin treatments? It is not necessary to have Table 1 and Figure 1 since they show the same data. It would be better to show Figure 1 only.
Lines 257, 258 Add error range to inhibition rates.
Figure 6 legend. Add length of incubation of the plates with infected pieces of chestnut. Also explain what 'Model' means. In the photo 6A provided for review, the images are very small and it is not possible to distinguish any differences. It would be much better to have much larger images. In 6B, the meaning of 'incidence' needs to be explained here, and in the Methods section (section 2.4.1?).
Discussion. One important point is whether the levels of natamycin used in this study are within the range permitted as a food additive. This should be discussed in the Discussion. The data shown in Figure 7 indicate that some chestnuts rotted regardless of treatments. How does this fit with commercial of a food? Was the decay in the control group also caused by the same fungus N. parvum, present within the fruit at the start of the experiment from natural infection (or from endophytic growth)?
Line 350 - 360. I do not understand why the authors discuss effects of natamycin on the cell wall, since its mode of action is via interaction with ergosterol in the cell membrane (e.g. DSM company website https://www.dsm.com/food-beverage/en_US/insights/insights/dairy/biopreservative-natamycin.html#:~:text=Natamycin%20is%20a%20natural%20preservative%20without%20safety%20risk&text=In%20the%20quantities%20applied%20to,confirmed%20by%20EFSA%20and%20FDA.). If they are proposing that there is actually a different mode of action, they need to provide much stronger data.
Minor points
Might be useful to have a sentence or two in the first paragraph that makes clear that chestnut is the name of the tree, and chestnut is also the name of its fruit (nuts). Could add 'tree' or 'fruit' if at any point writing about the tree but it is not obvious whether the tree or fruit is meant.
All systematic names should be in italics (e.g. line 44, Streptomyces scabies; line 130, 161, 171, 246 N. parvum and elsewhere).
Line 46 Dextran Sulfate Sodium Salt No need for capitals.
Line 52 N. parvum should be the name Neofusicoccum parvum in full since it is the first time it is mentioned in the paper (abstract does not count).
Line 87 More conventional to say Zhou (or Liu) and colleagues, rather than Dr Gao Zhou
Line 127 Define AKP
Line 178 Probably 'safety razor blade' rather than 'security blade'?
Line 178 Probably use 'test group' rather than 'model group'?
Line 179 (and elsewhere) Probably use 'control group' rather than 'blank group'?
Line 185 Corruption (here and elsewhere) is not the correct word. Infection?
Line 187 What does shell bucket mean?
Line 195 Stick, not sick
Line 319 'antifungal medication' is not the correct term. I think the authors mean it is the only antifungal additive for food?
Figure 7 axis labelled 'Statistical Times'. What does this mean?
Reviewer 3 Report
Paper reports attempt to use to use natamycin, widely used polyolefin macrolide antifungal compound, against N. parvum as the pathogenic fungus infecting chestnut fruits. They established natamycin MIC and MFC and used several methods to study possible mechanism of action against this particular type of fungi both in vitro and in vivo.
This is simple article on biological testing of one antibiotic against one type of species. As authors pointed out these are preliminary studies and as such methodology level is correct and results are critically concluded at the end. Since use of natamycin in the above context is new, article has some scientific value and it is worth to read.
However, article needs several improvements as shown below.
1. Natamycin is active against most fungi and yeast at low concentrations. The minimum inhibitory concentration (MIC) for natamycin against all foodborne fungi and yeast is less than 20 ppm. Natamycin is allowed for external use in humans and in diary industry for example for stabilization of the amount of fungi on the surface of cheese products. WHO restricts amount of this antibiotic to maximum 1 mg/dm2. Usually producers recommend also to remove external skin of cheese before cheese consumption. The same concerns meat products. Although most often chestnut fruits are thermally treated before consumption prior contact with human skin and therefore with mucosa, is highly probable. Effective MIC and MFC against N. parvum found in these studies are relatively high. Can authors comment on this in Introduction part and in Conclusions?
2. Moreover, natamycin is relatively chemically unstable and probably should not be used for longer storage of chestnut products.
3. Page 255: “N.parvum, which was isolated from chestnut, belongs to Ascomycota, Dothideomycetes, Botryosphaeriales, Botryosphaeriaceae, and Botryosphaeria,which are new species of asexual fungi...”
I am not familiar with classification of fungal genera. Therefore, I am asking how this is possible that N Parvum belongs to five “families”? Why some of them are written in italics and some are not?
4. Page 2/86 : “N.parvum was isolated from infected chestnuts and identified using the morphological and molecular biology methods by Dr. Gao Zhou[28].”
Please give brief description of the procedure to be sure that this is the only type of fungi present in the studied crop of chestnuts.
5. Page 10/377: “In vivo experiments showed that natamycin emulsion could effectively prolong the storage period of chestnuts and could be used as a green preservative …”
Although most of antibiotics are of natural origin it is not alllowed to call them “green”. This might in the future result in their misuse or overdosing as in the case of pecicillins. Please rephrase.
Summing up, article might be published after minor evision.
Round 2
Reviewer 2 Report
This manuscript is now substantially improved. It explains the rationale behind the study, the methodology and the results much more effectively.
Two minor points:
The sentence at line 79 - 84 about natamycin's mode of action is too long and difficult to understand. Can it be split into shorter sentences?
The term shell bucket is unclear. Can it be defined somewhere, such as line 99?